# Microbiological, Physicochemical and Sensorial Changes during the Ripening of Sucuk, a Traditional Turkish Dry-Fermented Sausage: Effects of Autochthonous Strains, Sheep Tail Fat and Ripening Rate

**Ahmet Akköse** [1] , **Şeyma Şişik Oğraş** [1] , **Mükerrem Kaya** [1,2] **and Güzin Kaban** [1,*]

1 Department of Food Engineering, Faculty of Agriculture, Atatürk University, TR-25240 Erzurum, Türkiye; akkose@atauni.edu.tr (A.A.); seymasisik@atauni.edu.tr (Ş.Ş.O.); mkaya@atauni.edu.tr (M.K.)
2 MK Consulting, Ata Teknokent, TR-25240 Erzurum, Türkiye
* Correspondence: gkaban@atauni.edu.tr

**Abstract:** This study aimed to investigate the effects of autochthonous starter cultures (spontaneous fermentation, *Lactiplantibacillus plantarum* GM77, *Staphylococcus xylosus* GM92 or *L. plantarum* GM77 + *S. xylosus* GM92) isolated from sucuk (a traditional Turkish dry fermented sausage), the use of sheep tail fat (beef fat-control, sheep tail fat and beef fat + sheep tail fat) and the ripening rate (slow or fast) on the microbiological, physicochemical and sensorial changes during the ripening of sucuk. *L. plantarum* GM77 as a monoculture or mixed culture with *S. xylosus* GM92 exhibited good growth during fermentation and following days of ripening. *S. xylosus* GM92 remained at the inoculation level of $10^6$ CFU/g. *L. plantarum* GM77 as a monoculture inhibited the growth of spontaneous *Micrococcus/Staphylococcus* in both the slow and fast ripening conditions. In the presence of *L. plantarum* GM77, the pH value decreased under 5.0 after the first three days of fermentation. The fast ripening yielded a lower mean $a_w$ and TBARS values compared to the slow ripening. Regarding TBARS value, the lowest mean value was determined in the presence of *L. plantarum* GM77 + *S. xylosus* GM92. The use of sheep tail fat caused an increase in TBARS; the highest mean value was determined in sucuk prepared with only sheep tail fat. The groups with *L. plantarum* GM77 yielded a higher mean L* value, while the highest a* value was determined in the group with *L. plantarum* GM77 + *S. xylosus* GM92. In addition, fast ripening caused an increase in the a* value. The L*, a* and b* values were not affected by the use of sheep tail fat. *L. plantarum* GM77 + *S. xylosus* GM92 groups demonstrated the best results in terms of general acceptability in both slow and fast ripening.

**Keywords:** sausage; sucuk; sheep tail fat; *Lactiplantibacillus plantarum*; *Staphylococcus xylosus*

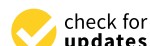


## 1. Introduction

Fermented meat products provide a significant variety of meat products due to their unique organoleptic properties [1]. Traditional dry fermented sausages are based on natural contamination by indigenous microflora, which may originate from the raw material used or from the environment. This microbiota, known as house flora, may contain microorganisms such as *Enterobacteriaceae*, *Pseudomonas* and *Enterococcus* that may adversely affect sucuk's product properties, pathogenic microorganisms, such as *Staphylococcus aureus* and *Listeria monocytogenes*, and technologically important microorganisms during ripening [2,3]. However, the use of starter culture is required for controlled fermentation.

Sucuk is a type of fermented sausage widely produced in Türkiye. Beef meat and beef fat are usually used in its production. Sheep tail fat can also be included in the formulation [1,4]. Sucuk batters filled in beef small intestines or collagen casings can be subjected to slow or fast fermentation depending on the initial fermentation temperature [1]. In sucuk, *Lactiplantibacillus plantarum* [5,6] and *Latilactobacillus sakei* [7] are by far

the most often isolated species among lactic acid bacteria. Other species isolated from su-cuk were *Lactiplantibacillus pentosus*, *Latilactobacillus curvatus*, *Limosilactobacillus fermentum*, *Levilactobacillus brevis*, *Lactococcus lactis ssp. lactis*, *Lactobacillus delbrueckii*, *Lacticaseibacillus rhamnosus*, *Pediococcus pentosaceus*, *Pediococcus acidilactici*, *Leuconostoc mesenteroides ssp. mesenteroides/dextranicum* and *Leuconostoc lactis* [5–9]. Gram-positive and catalase-positive cocci is another important group of traditional sucuk microbiota. Among these microorganisms, staphylococci constitute the most important part, and *Staphylococcus xylosus* and *S. saprophyticus* are the most dominant species in sucuk [6]. A limited number of studies have been conducted on the possibilities of using lactic acid bacteria and coagulase negative staphylococci isolated from sucuk as starter cultures [10–12]. In addition, no study has been found to determine the effects of autochthonous strains, ripening rate and sheep tail fat on the product properties of sucuk during ripening. The present study aimed to determine the effects of the ripening rate, autochthonous starter culture, sheep tail fat and ripening time on the microbiological and physicochemical properties of sucuk. Microbiological (lactic acid bacteria, *Micrococcus/Staphylococcus* and Enterobacteriaceae) and physicochemical (pH, $a_w$, TBARS and color) analyses were performed on sucuk samples on certain days of ripening. In addition, the sensory properties of ripened sausages were also investigated.

## 2. Materials and Methods

### 2.1. Materials

Beef meat was taken from the shoulder of carcasses conditioned at 4 °C ± 1 °C for 24 h and obtained from a local slaughterhouse (Meat and Milk Institution, Erzurum, Türkiye). Beef fat and sheep tail fat were also taken from the Erzurum Meat and Dairy Institution's slaughterhouse. The raw materials were cut into small pieces after trimming and stored at −20 °C until sucuk production.

### 2.2. Sucuk Production

The sucuk formulation specified by Kaya and Gökalp [13] was used in the production process. It consisted of 80% beef meat + 20% fat, 2.5% NaCl, 1% garlic, 0.4% sucrose, 0.25% pimento, 0.9% cumin, 0.7% red pepper and 0.5% black pepper. For each treatment, 4 kg of beef + 1 kg of fat were used. Two productions were completed for each treatment; thus, 48 batters were produced according to the experimental design (given in Supplementary Material Table S1). The initial fermentation temperature was 18 °C for slow ripening and 24 °C for rapid ripening. Therefore, nitrate (150 mg/kg) for slow-ripened sausages and nitrite (150 mg/kg) for fast-ripened sausages were included as curing agents in the formulation.

*Lactiplantibacillus plantarum* GM77 (at the level of about $10^7$ CFU/g) and *Staphylococcus xylosus* GM92 (at the level of about $10^6$ CFU/g) strains isolated and identified from sucuk by Kaban and Kaya [6] were used as starter cultures. The batters without starter culture were evaluated as control group. A laboratory-type bowl cutter (MADO Typ MTK 662, Dornhan, Schwarzwald) was used in the preparation of sucuk batters. Prepared batters were filled into casings (38 mm, collagen material, Naturin Darm, Germany) using a laboratory-type piston filling machine (MADO Typ MTK 591, Dornhan, Schwarzwald). After filling, the samples were placed in the climate chamber (Reich, Urbach, Germany), where the temperature and relative humidity could be adjusted automatically. Depending on the experimental plan, two different initial fermentation temperatures were applied: In the slow ripening program, 18 ± 1 °C was applied between days 1 and 10 and 16 ± 1 °C between days 11 and 14; in the fast ripening program, 24 ± 1 °C was applied on the first day, 22 ± 1 °C on the second and third days, 20 ± 1 °C between days 4 and 6, 18 ± 1 °C between days 7 and 10 and 16 ± 1 °C between days 11 and 14. The relative humidity was gradually decreased from 92 ± 2% to 84 ± 2% in both ripening processes. The air flow rate, on the other hand, was gradually reduced from fast to slow.

### 2.3. Sampling Procedure

For each production, two samples were taken from each treatment. The analyzes were carried out on samples taken after 0, 1, 3, 7, 11 and 14 days of ripening. The sensory assessment was carried out on the 14th day of ripening (final product).

### 2.4. Microbiological Analyses

The sausage casing was removed aseptically. 25-g samples from each sausage were transferred to a sterile stomacher bag, and 225 mL of sterile physiological water (0.855 NaCl) were added. Then, the sample was homogenized for 1.5 min in a stomacher (Lab Blender 400-BA 7021, London, UK), and serial decimal dilutions were prepared.

Lactic acid bacteria were enumerated on de Man Rogosa Sharpe Agar (MRS, Merck, Darmstadt, Germany) in anaerobic conditions (Anaerocult A, Merck, Darmstadt, Germany) after 48 h at 30 °C; *Micrococcus/Staphylococcus* on Mannitol Salt Phenol Red Agar (MSA, Merck, Darmstadt, Germany) after 48 h at 30 °C. Enterobacteriaceae were enumerated on Violet Red Bile Agar (VRBD, Merck, Darmstadt, Germany) in anaerobic conditions (Anaerocult A, Merck, Darmstadt, Germany) after 48 h at 30 °C [14].

### 2.5. Physicochemical Analysis

The $a_w$ value of the samples was determined using the $a_w$ instrument (TH-500 $a_w$ Sprint, Novasina, Pfaffikon, Switzerland). The samples were placed in the measuring chamber at 25 °C. For the pH analysis, 10 g of homogenized samples were mixed with 100 mL of distilled water and then homogenized with ultra-turrax (IKA T25, IKA Werke GmbH & Co, Breisgau, Germany) for 1 min. The pH values were determined using a pH-meter (ATI Orion 420, Orion Inc., Boston, MA, USA). The method given by Lemon [15] was used to determine TBARS, and the results were given in μmol MDA/kg. The color values (L* (lightness), a* (red-green component) and b* (yellow-blue component)) in the samples sliced in 1 cm thickness were measured using a Chroma meter (Minolta, Osaka, Japan) with a D65 illuminant, an aperture size of 8 mm and a 2° standard observer. The measurements were performed in triplicate. The calibration was performed on a white standard plate (Y = 87.4, x = 0.3184, y = 0.3364) to standardize the equipment before each measurement [14].

### 2.6. Sensory Analysis

The sensory evaluation was performed on final products. A structured nine-point hedonic-type scale was used for analysis, and the color, texture, odor, taste and general acceptability of the samples were evaluated. A total of 20 semi-trained panelists, consisting of 10 females and 10 males, who are academic staff at the Department of Food Engineering (Atatürk University, Erzurum, Türkiye) participated in the sensory evaluation. The panelists were informed about sampling, analyzing and interpreting stimuli and providing responses based on relevant scores prior to analysis. In addition, they informed in detail the scorecard and the evaluation procedures. Samples sliced 1 cm thick were used in the analysis, and a portion of each sample unit was randomly selected. The analysis was performed at room temperature (22 °C ± 2 °C), under uniform lighting in a quiet laboratory. The panelists were told to take water and bread between samples and wait at least 30 s between samples to clear their palate. The analyses were performed in two different sessions and on two different days. In total, 40 sensory analyses results were obtained for each parameter.

### 2.7. Statistical Analysis

The experiment was carried out according to a completely randomized design with two replicates (two batters for each treatment). Two independent batters of sausages (replicates) were prepared on different days. The ripening rate, sheep tail fat, starter culture and ripening time were evaluated as the main effects, and replications were evaluated as the random effects. The analysis of variance was applied to the data. The means of significant sources of variation were compared using Duncan's multiple range tests at the

$p < 0.05$ level. The results were expressed as the mean values $\pm$ standard deviation (Sd). All statistical analysis was performed using the SPSS statistical program (IBM SPSS Inc., Chicago, IL, USA).

## 3. Results and Discussion

### 3.1. Lactic Acid Bacteria

The overall effects of the ripening rate, autochthonous starter culture, sheep tail fat and ripening time on the microbiological and physicochemical changes of sucuk are shown in Table 1. The ripening rate, sheep tail fat, autochthonous starter culture and ripening time factors had a significant effect on the lactic acid bacteria count at the $p < 0.01$ level. The lowest mean value in terms of the ripening rate was determined in fast ripening (Table 1). However, while a 0.5-log unit increase in the number of lactic acid bacteria on day 1 was observed in slow ripening, this increase was about 1 log unit in fast ripening. Higher lactic acid bacteria count was determined on the third day of slow ripening (Figure 1a). This result is probably due to the faster initial pH drop in fast ripening. Higher lactic acid bacteria counts were also detected in slow-ripened sucuk on other days of ripening. In addition, these results demonstrate that *L. plantarum* GM77 has adapted well to both ripening conditions, and that the number of lactic acid bacteria remained at the level of $10^8$ CFU/g (Figure 1a). Soyer et al. [16] stated that the interaction of fermentation temperature $\times$ ripening time has a very important effect on the number of lactic acid bacteria in sucuk.

Statistical differences were determined between the group containing lactic acid bacteria alone and the other groups. The groups with spontaneous microbiota (control) and *S. xylosus* GM92 yielded the lowest values in terms of lactic acid bacteria count. There were significant differences between the average lactic acid bacteria counts of groups with starter culture and without starter culture. Only the group containing *L. plantarum* GM77 showed the highest mean lactic acid bacteria count. Further, the *L. plantarum* GM77+ *S. xylosus* GM92 group had a higher mean value than the control and *S. xylosus* groups (Table 1). These results demonstrate that *L. plantarum* GM77 alone and together with *S. xylosus* GM92 exhibits good growth in sucuk. In addition, *L. plantarum* GM77 exhibited good growth in the fermentation stage and continued its survival in the following days of ripening (Figure 1b). Similar results were also determined by Kaban and Kaya [10,11], who used the same strain (*L. plantarum* GM77) as a starter culture in sucuk production. On the other hand, some differences were observed between the fat groups during the ripening of the sucuk samples (Figure 2c).

**Table 1.** The overall effects of ripening rate, sheep tail fat, autochthonous starter culture and ripening time on the microbiological and physicochemical changes of sucuk (mean $\pm$ sd).

| Factors | N | Lactic Acid Bacteria (CFU/g) | *Micrococcus/ Staphylococcus* (CFU/g) | pH | $a_w$ | TBARS (μmol MDA/kg) |
|---|---|---|---|---|---|---|
| **Ripening rate (RR)** | | | | | | |
| Slow (18 °C) | 144 | 7.35 $\pm$ 1.48 a | 6.32 $\pm$ 1.22 a | 5.29 $\pm$ 0.48 a | 0.914 $\pm$ 0.044 a | 7.15 $\pm$ 2.94 a |
| Fast (24 °C) | 144 | 7.29 $\pm$ 1.30 b | 6.05 $\pm$ 1.30 b | 5.30 $\pm$ 0.45 a | 0.891 $\pm$ 0.055 b | 5.95 $\pm$ 2.29 b |
| Significance | | ** | ** | ns | ** | ** |
| **Sheep tail fat (STF)** | | | | | | |
| Beef fat- control | 96 | 7.41 $\pm$ 1.30 a | 6.23 $\pm$ 1.24 a | 5.30 $\pm$ 0.45 a | 0.902 $\pm$ 0.052 b | 5.66 $\pm$ 1.32 c |
| Sheep tail fat | 96 | 7.23 $\pm$ 1.50 c | 6.17 $\pm$ 1.31 b | 5.29 $\pm$ 0.46 a | 0.903 $\pm$ 0.051 a | 7.68 $\pm$ 3.80 a |
| Beef fat + sheep tail fat | 96 | 7.33 $\pm$ 1.38 b | 6.14 $\pm$ 1.25 b | 5.30 $\pm$ 0.48 a | 0.902 $\pm$ 0.052 b | 6.30 $\pm$ 1.92 b |
| Significance | | ** | ** | ns | * | ** |

**Table 1.** *Cont.*

| Factors | N | Lactic Acid Bacteria (CFU/g) | *Micrococcus/ Staphylococcus* (CFU/g) | pH | a$_W$ | TBARS (µmol MDA/kg) |
|---|---|---|---|---|---|---|
| **Starter culture (SC)** | | | | | | |
| Control without starter | 72 | 6.62 ± 1.58 c | 5.91 ± 1.01 c | 5.55 ± 0.24 a | 0.905 ± 0.048 a | 7.89 ± 3.92 a |
| *L.plantarum* | 72 | 8.14 ± 0.66 a | 4.58 ± 0.67 d | 5.05 ± 0.50 b | 0.900 ± 0.054 b | 6.50 ± 2.42 b |
| *S.xylosus* | 72 | 6.61 ± 1.62 c | 7.46 ± 0.45 a | 5.55 ± 0.23 a | 0.905 ± 0.051 a | 6.29 ± 1.95 b |
| *L.plantarum + S. xylosus* | 72 | 7.91 ± 0.51 b | 6.77 ± 0.25 b | 5.04 ± 0.49 b | 0.899 ± 0.054 c | 5.51 ± 1.22 c |
| Significance | | ** | ** | ** | ** | ** |
| **Ripening time (RT) (day)** | | | | | | |
| 0 | 48 | 5.55 ± 1.55 d | 5.59 ± 1.20 f | 5.78 ± 0.04 a | 0.954 ± 0.002 a | 4.30 ± 0.39 e |
| 1 | 48 | 6.31 ± 1.57 c | 5.87 ± 1.28 e | 5.76 ± 0.15 b | 0.949 ± 0.004 b | 5.45 ± 0.75 d |
| 3 | 48 | 8.03 ± 0.48 a | 6.69 ± 0.94 a | 5.18 ± 0.46 c | 0.935 ± 0.011 c | 6.05 ± 1.45 c |
| 7 | 48 | 8.07 ± 0.37 a | 6.46 ± 1.14 b | 4.98 ± 0.35 e | 0.900 ± 0.024 d | 7.33 ± 2.62 b |
| 11 | 48 | 8.06 ± 0.42 a | 6.28 ± 1.36 c | 4.99 ± 0.34 e | 0.851 ± 0.022 e | 7.21 ± 2.05 b |
| 14 | 48 | 7.92 ± 0.44 b | 6.20 ± 1.35 d | 5.09 ± 0.34 d | 0.825 ± 0.015 f | 8.95 ± 4.13 a |
| Significance | | ** | ** | ** | ** | ** |
| **Interactions** | | | | | | |
| RR × STF | | ns | ns | ** | * | ** |
| RR × SC | | ns | ** | ** | ns | ** |
| RR × RT | | ** | ** | ** | ** | ** |
| STF × SC | | ns | ns | ** | ns | ** |
| STF × RT | | ** | ns | ns | ** | ** |
| SC × RT | | ** | ** | ** | ** | ** |
| RR × STF × SC | | ns | * | ns | ns | ** |
| RR × STF × RT | | ns | ns | ns | * | ** |
| RR × SC × RT | | * | ** | ** | ** | ** |
| STF × SC × RT | | ** | ns | ns | ns | ** |
| RR × STF × SC × RT | | ns | ns | ns | ns | ** |

sd: standard deviation: ns: not significant; a–f: any two means in the same column having the same letters in the same section are not significantly different at $p > 0.05$, ** $p < 0.01$, * $p < 0.05$.

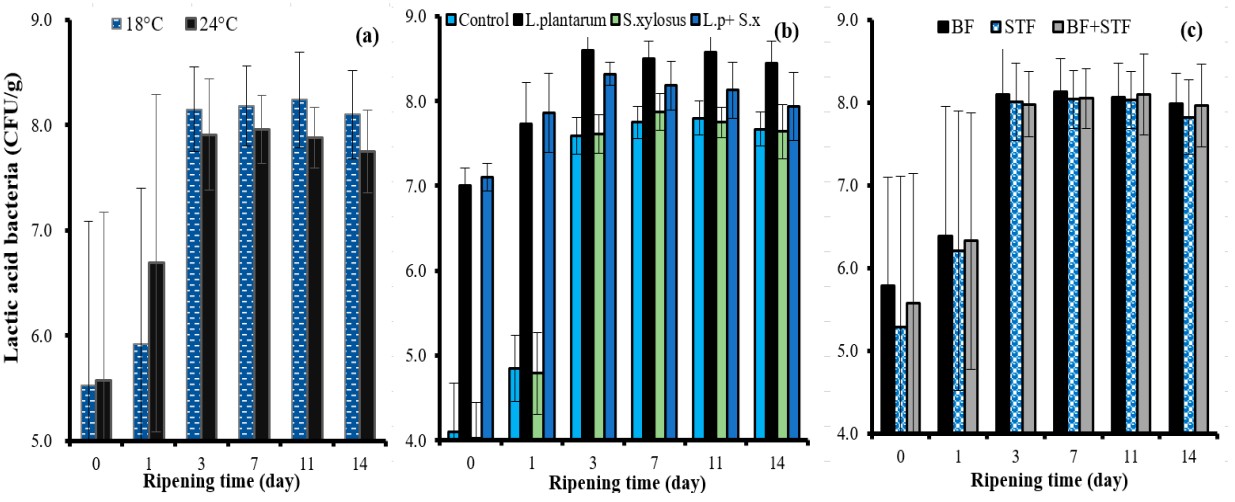

**Figure 1.** The effects of the interactions of ripening rate × ripening time (**a**); starter culture × ripening time (**b**); and sheep tail fat × ripening time (**c**) on the lactic acid bacteria count of sucuk.

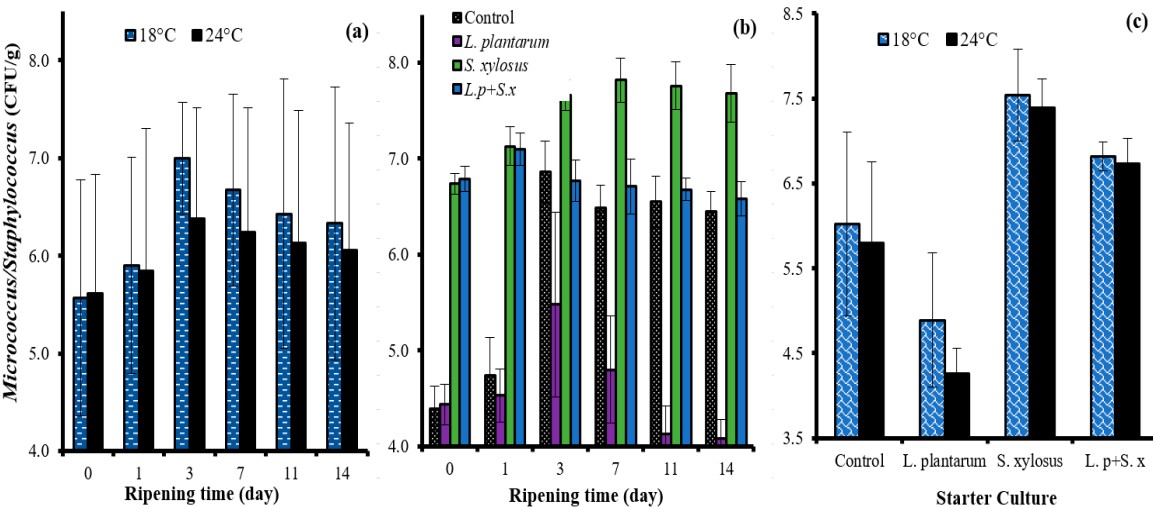

**Figure 2.** The effects of the interactions of ripening rate × ripening time (**a**); starter culture × ripening time (**b**); and ripening rate × starter culture (**c**) on the *Micrococcus/Staphylococcus* count of sucuk.

### 3.2. Micrococcus/Staphylococcus

The most important stage for the development of micrococci and staphylococci comprises the first days of ripening. The growth of these acid-sensitive bacteria is significantly affected by rapid acidification. Therefore, the lowest mean value was determined in fast-ripening (24 °C) sucuk in the study (Table 1). In previous studies on sucuk, it was also determined that coagulase-negative staphylococci were significantly affected by acidification [10,11,17,18].

*Micrococcus/Staphylococcus* exhibited similar growth on the first day of slow and fast ripening, while faster growth occurred on the next two days at 18 °C. After the third day of ripening, both initial fermentation temperature applications caused a decrease in the number (Figure 2a). As can be seen from Table 1, the mean lowest number for *Micrococcus/Staphylococcus* was observed in the group containing only *L. plantarum* GM77 due to the rapid drop in pH value. The highest mean value was determined only in the group with *S. xylosus* GM92. The factor of sheep tail fat also had an effect on the number of *Micrococcus/Staphylococcus*, and the highest mean value was determined in the beef fat group (Table 1). On the other hand, as can be seen in Figure 2b, *S. xylosus* GM92 exhibited good adaptation to the environment even in the presence of *L. plantarum* GM77. Furthermore, between the *S. xylosus* GM92 and *L. plantarum* GM77 + *S. xylosus* GM92 groups, the initial fermentation temperature did not have a significant effect on the number of *Micrococcus/Staphylococcus* (Figure 2c).

These results indicated that *S. xylosus* GM92 can be used as a starter culture in both slow- and fast-ripened sucuk. On the other hand, when *L. plantarum* GM77 was used alone, it caused a decrease in the number, especially in fast ripening. These results indicate that *L. plantarum* GM77 negatively affects the growth of coagulase-negative staphylococci, which have important functions in the formation of color and aroma, and therefore *L. plantarum* GM77 should be added to the sucuk batter together with *S. xylosus* GM92 in both fast- and slow-ripening sucuk.

### 3.3. Enterobacteriaceae

In this study, the Enterobacteriaceae count was determined below $10^2$ CFU/g both at the beginning of ripening (day 0) and on the other analysis days. Since the members of this family are microorganisms sensitive to low pH (<5.3) and $a_w$ (<0.960) values, their counts decrease depending on the decreasing pH and $a_w$ values during ripening. Similar results have been determined by other researchers [11,17,19].

### 3.4. pH

The ripening rate had no significant effect on the mean pH value of the sucuk ($p > 0.05$). However, the interaction of ripening rate × ripening time affected the pH value of the sucuk (Table 1). As seen in Figure 3a, in rapid ripening, the pH started to decrease after 24 h of fermentation, and this decrease continued until the 7th day. The most significant decrease was observed on the third day of ripening. On the other hand, in slow ripening, a slight increase in the pH value was observed on the first day, and the pH value decreased until the 11th day; however, the most significant decrease occurred on the third day of ripening. It was thought that the slight increase in pH on the first day of ripening in slow ripening was probably due to the slow activity of lactic acid bacteria and, accordingly, the metabolites formed as a result of the activity of other bacteria in the sucuk batter. Similarly, Kaban and Kaya [18] and Kaban and Kaya [10] stated that there was a slight increase in the pH value after 24 h of fermentation in sucuk produced through natural fermentation.

The use of sheep tail fat did not have a significant effect on the pH value of the sucuk. On the other hand, the use of starter culture had a very important effect. The control group and the group using *S. xylosus* GM92 yielded a higher mean pH value than the groups containing *L. plantarum* GM77 (Table 1). It was also determined by other researchers that the use of lactic starter cultures in sucuk and similar fermented products significantly decreases the pH value [10,20]. In this study, as seen in Figure 3b, the pH rapidly decreased in the presence of *L. plantarum* GM77 and dropped below 5.0 on the third day. In the control and *S. xylosus* GM92 groups, a slight increase was observed at the beginning (day 1), and then a decrease was observed. During ripening, the groups containing *L. plantarum* GM77 had lower pH values than the other groups (Figure 3b). In many studies on sucuk, it has been also determined that the pH drops below 5.0 in the first 3 days when lactic starter cultures are used in production [10,11,20].

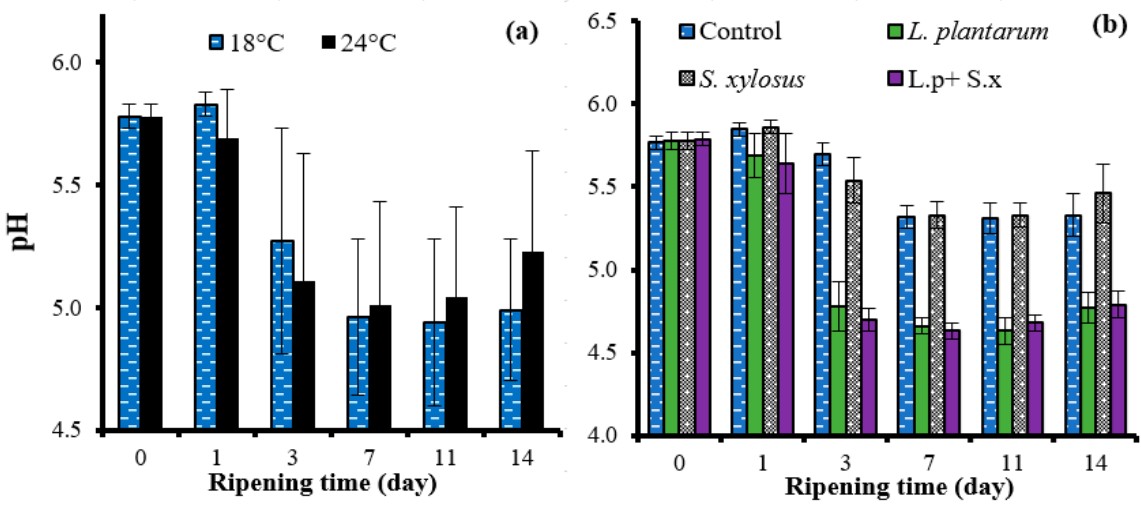

**Figure 3.** The effects of interactions of ripening rate × ripening time (**a**) and starter culture × ripening time (**b**) on the pH of sucuk.

### 3.5. Water Activity ($a_w$)

The fast-ripening temperature yielded a lower average $a_w$ value than the slow-ripening temperature (Table 1). This result is because the pH decreases faster in the initial days of ripening compared to slow ripening in fast ripening, and accordingly drying is faster. On the other hand, the *L. plantarum* GM77 + *S. xylosus* GM92 group had the lowest mean $a_w$ value. In contrast, the control and *S. xylosus* GM92 group yielded the highest mean $a_w$ values (Table 1).

It has been determined in other studies on sucuk that $a_w$ decreases faster in sucuk produced using lactic starter culture than in sucuk naturally fermented [10,17,18]. The $a_w$ value of the sucuk decreased as the time progressed (Table 1). However, this decrease was quicker in fast ripening (Figure 4a). Lactic acid bacteria are effective in fermented sausages due to their acid formation and therefore also have an impact on drying. In the present study, lower $a_w$ values were generally detected in the final product in the presence of *L. plantarum* GM77 (Figure 4b). The use of only sheep tail fat also caused some differences in the $a_w$ value during ripening (Table 1, Figure 4c).

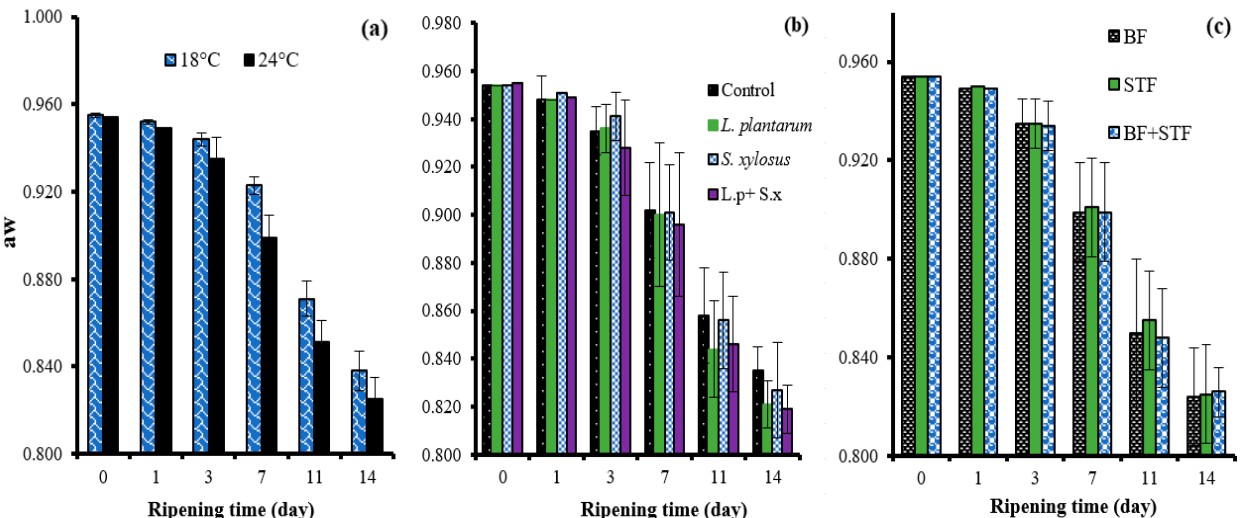

**Figure 4.** The effects of the interactions of ripening rate and ripening time (**a**); starter culture and ripening time (**b**); and sheep tail fat and ripening time (**c**) on the $a_w$ of sucuk.

*3.6. Thiobarbituric Acid Reactive Substances (TBARS)*

The TBARS value, an indicator of lipid oxidation, yielded a higher mean TBARS value in slow ripening than in fast ripening (Table 1). However, as seen in Figure 5a, the TBARS value in the final product (14 days) was found below 10 μmol MDA/kg in both fast and slow ripening. On the other hand, the group containing 20% sheep tail fat had a higher mean value than the group containing 10% beef fat + 10% sheep tail fat, while the lowest mean value was determined to be in the group containing 20% beef fat (Table 1). During ripening, sheep tail fat yielded higher values compared to other groups (control and beef fat + sheep tail fat)(Figure 5b). These results were because sheep tail fat is more sensitive to oxidation than beef fat [21].

The use of starter culture in sucuk decreased the TBARS value. The *L. plantarum* GM77+ *S. xylosus* GM92 group yielded a lower TBARS value than the other monoculture groups (*L. plantarum* GM77 and *S. xylosus* GM92). Kaban et al. [12] reported that sucuk produced with *S. carnosus* G109 + *L. sakei* S15 or *S. carnosus* G109 + *L. sakei* S15 + *L. plantarum* S91 have a lower TBARS value than the control group and the groups to which only autochthonous lactic acid bacteria were added. They also stated that this result may be due to the antioxidant activity of *S. carnosus* G109.

As the ripening time progressed, the TBARS value increased (Table 1). Another important result of the TBARS values of samples was that the *L. plantarum* + *S. xylosus* culture yielded lower values from the third day of ripening compared to the other groups (control, *L. plantarum*, *S. xylosus*) (Figure 5c). It was also determined that the TBARS value increased as the ripening time progressed in other studies [10,11,16].

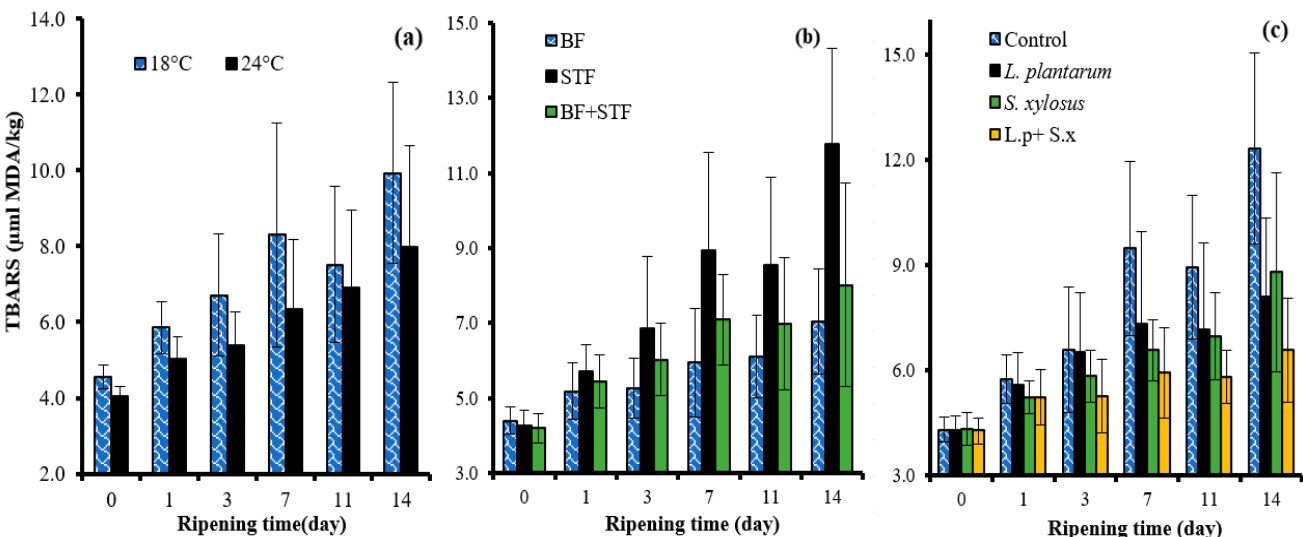

**Figure 5.** The effects of the interactions of ripening rate × ripening time (**a**); sheep tail fat × ripening time (**b**); and starter culture × ripening time (**c**) on the TBARS value of sucuk.

*3.7. Color*

The L* value, a measure of brightness, was higher in the *L. plantarum* GM77 groups than in the control and *S. xylosus* GM92 groups. The mean L* value increased as the ripening period progressed, but there was no statistical difference between the values from the seventh day (Table 2). Starter cultures exhibited different effects during ripening. *L. plantarum* GM77 and *L. plantarum* GM77+ *S. xylosus* GM92 caused a significant increase in L* on day 3 of ripening. Similarly, it has been reported that lactic acid bacteria also increase the L* value in dry cured meat products [22]. In this study, there was generally no significant change during ripening in the control group. In the *S. xylosus* GM92 group, the L* value remained below 40 during ripening. According to these results, *S. xylosus* GM92 alone did not cause a significant increase in the L* value, an indicator of lightness-darkness (Figure 6a).

The a* value, which expresses the red color intensity, yielded a higher mean value in fast ripening than in slow ripening (Table 2). Soyer et al. [16] stated that color conversion was better at 24–26 °C fermentation temperature than 20–22 °C. In the present study, the highest mean a* value was determined in a mixed culture (*L. plantarum* GM77 + *S. xylosus* GM92). The lowest mean value was observed in the *L. plantarum* GM77 group, but this mean was not different from that of the control group. Moreover, there were changes depending on the ripening time: the a* value increased until the third day and decreased in the following days (Table 1). This result demonstrates that the first 3 days are very important in terms of color formation in sucuk. However, the effects of starter cultures also differed during ripening. On the first day of ripening, the a* value increased in all groups except the control group. On the third day of ripening, the a* value increased in all groups including the control group. On the seventh day of ripening, there was an increase only in the *L. plantarum* GM77 + *S. xylosus* GM92 group and a decrease in the other groups. Although the a* value decreased in all groups at the last stages of ripening, the *L. plantarum* GM77+ *S. xylosus* GM92 group exhibited the highest values at this stage (Figure 6b). This result demonstrates that color stability is better in the later stages of ripening in the presence of *L. plantarum* GM77 + *S. xylosus* GM92. In the light of these evaluations, it can be concluded that the *L. plantarum* GM77 + *S. xylosus* GM92 mixed culture is the most suitable culture in terms of its a* value, which is an indicator of red color intensity. Kaban and Kaya [23] corroborated that this mixed culture had better color results than the control.

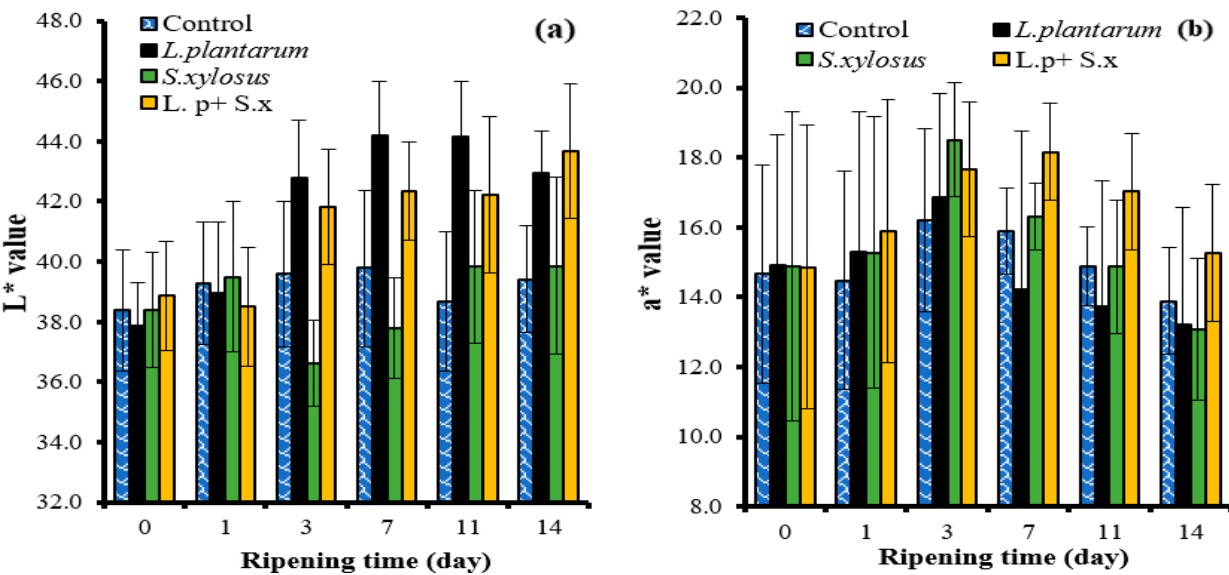

**Figure 6.** The effects of the interactions of starter culture × ripening time on the L* value (**a**) and the a* value (**b**) of sucuk.

**Table 2.** The overall effects of ripening rate, sheep tail fat, autochthonous starter culture and ripening time on the L*, a* and b* values of sucuk (mean ± sd).

| Factors | N | L* | a* | b* |
|---|---|---|---|---|
| **Ripening rate (RR)** | | | | |
| Slow (18 °C) | 144 | 40.28 ± 2.95 a | 14.56 ± 2.97 b | 12.33 ± 5.96 a |
| Fast (24 °C) | 144 | 40.17 ± 2.95 a | 16.28 ± 3.07 a | 7.49 ± 2.41 a |
| Significance | | ns | ** | ns |
| **Sheep tail fat (STF)** | | | | |
| Beef fat- control | 96 | 40.21 ± 2.93 a | 15.47 ± 2.98 a | 7.81 ± 2.25 a |
| Sheep tail fat | 96 | 39.99 ± 3.17 a | 15.34 ± 3.29 a | 7.69 ± 2.39 a |
| Beef fat + sheep tail fat | 96 | 40.47 ± 2.72 a | 15.44 ± 3.15 a | 14.23 ± 6.42 a |
| Significance | | ns | ns | ns |
| **Starter culture (SC)** | | | | |
| Control without starter | 72 | 39.18 ± 2.19 b | 15.00 ± 2.36 c | 7.44 ± 2.37 a |
| *L.plantarum* | 72 | 41.81 ± 3.06 a | 14.71 ± 3.77 c | 8.03 ± 2.34 a |
| *S.xylosus* | 72 | 38.66 ± 2.46 b | 15.49 ± 3.14 b | 7.64 ± 2.47 a |
| *L.plantarum + S. xylosus* | 72 | 41.24 ± 2.76 a | 16.48 ± 2.85 a | 16.52 ± 7.04 a |
| Significance | | ** | ** | ns |
| **Ripening time (RT) (day)** | | | | |
| 0 | 48 | 38.36 ± 1.79 c | 14.83 ± 3.74 c | 11.49 ± 1.05 a |
| 1 | 48 | 39.06 ± 2.18 c | 15.24 ± 3.62 c | 9.19 ± 1.31 a |
| 3 | 48 | 40.20 ± 3.06 b | 17.31 ± 2.43 a | 8.24 ± 1.17 a |
| 7 | 48 | 41.03 ± 3.12 a | 16.14 ± 2.77 b | 6.73 ± 1.07 a |
| 11 | 48 | 41.22 ± 3.12 a | 15.13 ± 2.50 c | 5.89 ± 0.90 a |
| 14 | 48 | 41.47 ± 2.82 a | 13.87 ± 2.41 d | 17.93 ± 8.61 a |
| Significance | | ** | ** | ns |

**Table 2.** *Cont.*

| Factors | N | L* | a* | b* |
|---|---|---|---|---|
| **Interactions** | | | | |
| RR × STF | | * | ns | ns |
| RR × SC | | * | ** | ns |
| RR × RT | | ns | ** | ns |
| STF × SC | | ns | ns | ns |
| STF × RT | | * | ** | ns |
| SC × RT | | ns | ns | ns |
| RR × STF × SC | | ns | ns | ns |
| RR × STF × RT | | ns | ns | ns |
| RR × SC × RT | | ns | ** | ns |
| STF × SC × RT | | ns | ** | ns |
| RR × STF × SC × RT | | ns | ns | ns |

sd: standard deviation: ns: not significant; a–d: any two means in the same column having the same letters in the same section are not significantly different at $p > 0.05$, ** $p < 0.01$, * $p < 0.05$.

### 3.8. Sensorial Properties

The effects of the ripening rate, starter culture and sheep tail fat on the sensory properties of sucuk are shown in Table 3. The ripening rate did not have a significant effect on the color. The highest mean color value was given by the beef fat group (control), but this value did not differ statistically from the group using only sheep tail fat. This result demonstrates that there is no negative effect on color when sheep tail fat is used alone as fat. As can be seen in Table 3, the beef fat + sheep tail fat group had a lower value than the beef fat group. Besides, starter culture × ripening rate interaction also had a significant effect on the color of the sucuk. *L. plantarum* GM77 had higher color scores in fast ripening and *S. xylosus* GM92 in slow ripening compared to the other groups. The control group, on the other hand, scored higher in slow ripening than in fast ripening (Figure 7a).

In terms of texture, slow-ripened sucuk had a higher average value than fast-ripened sucuk. However, as can be seen in the starter culture × ripening rate interaction shown in Figure 7b, while the use of starter culture did not cause a significant change in the texture in slow ripening, *L. plantarum* GM77 and *L. plantarum* GM77+ *S. xylosus* GM92 significantly increased the texture scores in fast ripening (Figure 7b). This result demonstrates the importance of using lactic starter culture in fast ripening. Similar results were also determined for taste and odor sensory properties. However, the use of *L. plantarum* GM77 decreased the taste and odor values in slow ripening (Figure 7c,d). The use of starter culture also caused significant differences in terms of the general acceptability parameter. The *L. plantarum* GM77 and *L. plantarum* GM77 + *S. xylosus* GM92 groups yielded the best results in fast ripening, and the *S. xylosus* GM92 and *L. plantarum* GM77+ *S. xylosus* GM92 groups in slow ripening. As can be seen in Figure 7e, the scores of the *L. plantarum* GM77 + *S. xylosus* GM92 group in fast and slow ripening are quite close to each other. Kaban and Kaya [23] stated that the mixed culture of *L. plantarum* GM77 + *S. xylosus* GM92 received higher scores in terms of texture, taste, odor and overall acceptability than the sucuk produced by applying natural fermentation.

**Table 3.** The overall effects of ripening rate, autochthonous starter culture and sheep tail fat on the sensory parameters of sucuk (mean ± sd).

| Factors | N | Color | Texture | Taste | Odor | General Acceptability |
|---|---|---|---|---|---|---|
| **Ripening rate (RR)** | | | | | | |
| Slow (18 °C) | 24 | 7.12 ± 0.79 a | 6.99 ± 0.33 a | 6.81 ± 0.37 a | 6.86 ± 0.51 a | 7.07 ± 0.47 a |
| Fast (24 °C) | 24 | 6.97 ± 0.79 a | 6.75 ± 0.91 b | 6.84 ± 0.70 a | 6.81 ± 0.68 a | 6.74 ± 0.83 b |
| Significance | | ns | ** | ns | ns | ** |
| **Sheep tail fat (STF)** | | | | | | |
| Beef fat- control | 16 | 7.26 ± 0.63 a | 7.04 ± 0.50 a | 7.00 ± 0.42 a | 7.06 ± 0.46 a | 7.16 ± 0.47 a |
| Sheep tail fat | 16 | 7.01 ± 0.84 ab | 6.77 ± 0.85 b | 6.71 ± 0.68 a | 6.74 ± 0.64 b | 6.73 ± 0.82 b |
| Beef fat + sheep tail fat | 16 | 6.86 ± 0.86 b | 6.80 ± 0.68 b | 6.78 ± 0.54 a | 6.71 ± 0.64 b | 6.83 ± 0.67 b |
| Significance | | * | * | ns | * | ** |
| **Starter culture (SC)** | | | | | | |
| Control without starter | 12 | 6.82 ± 0.65 b | 6.53 ± 0.68 b | 6.66 ± 0.48 b | 6.55 ± 0.51 b | 6.63 ± 0.72 c |
| *L.plantarum* | 12 | 6.77 ± 0.91 b | 7.19 ± 0.45 a | 7.02 ±0.50 a | 6.85 ±0.58 b | 7.02 ±0.52 b |
| *S.xylosus* | 12 | 7.01 ± 0.83 b | 6.49 ± 0.77 b | 6.58 ± 0.65 b | 6.70 ± 0.57 b | 6.57 ± 0.79 c |
| *L.plantarum + S. xylosus* | 12 | 7.58 ± 0.48 a | 7.28 ± 0.45 a | 7.06 ± 0.48 a | 7.24 ± 0.56 a | 7.41 ± 0.31 a |
| Significance | | ** | ** | ** | ** | ** |
| **Interactions** | | | | | | |
| RR × STF | | ns | ns | ns | ns | ns |
| RR × SC | | ** | ** | ** | ** | ** |
| STF × SC | | ns | ns | ns | ns | ns |
| RR × STF × SC | | ns | ns | ns | ns | ns |

sd: standard deviation: ns: not significant; a–c: any two means in the same column having the same letters in the same section are not significantly different at $p > 0.05$; * $p < 0.05$; ** $p < 0.01$.

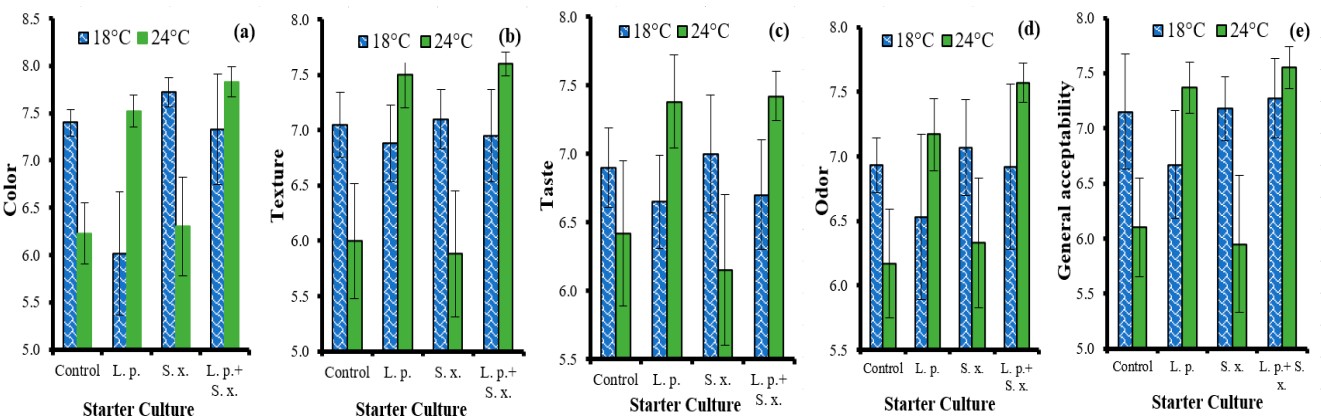

**Figure 7.** The effects of the interactions of ripening rate × starter culture on color (**a**); texture (**b**), taste (**c**); odor (**d**); and general acceptability (**e**) of sucuk.

## 4. Conclusions

In fermented sausages, the fermentation stage is an important step in developing the product characteristics. When the *L. plantarum* GM77 strain isolated from sucuk was used as a monoculture or mixed culture, it caused a significant decrease in the pH value in the first 3 days of fermentation. *S. xylosus* GM92 isolated from sucuk also remained viable during ripening when used alone or together with *L. plantarum* GM77 as a starter culture. However, in sucuk prepared using only *L. plantarum* GM77, spontaneous *Micrococcus/Staphylococcus* was significantly inhibited. Fast ripening yielded a lower $a_w$ value. The *L. plantarum* GM77

and *S. xylosus* GM92 mixed culture yielded the lowest average $a_w$ value. The use of sheep tail fat (with and without beef fat) caused an increase in the TBARS value of sucuk. On the other hand, lower TBARS values were determined at the end of ripening in the groups using starter culture, compared to the control. The brightness value, which is an important quality criterion in sucuk, increased with the use of *L. plantarum* GM77 (with or without *S. xylosus* GM92). The a* value increased with the use of mixed culture. Mixed culture demonstrated the best results in terms of sensory properties at both initial fermentation temperatures. On the other hand, it is also important to determine the effects of autochthonous strains on the volatile compounds and textural parameters of sucuk under different process conditions. In addition, the use of starter culture is essential for standard industrial production. In this respect, it is of great importance to transform autochthonous strains into commercial preparations.

**Supplementary Materials:** The following supporting information can be downloaded at: https://www.mdpi.com/article/10.3390/fermentation9060558/s1.

**Author Contributions:** Conceptualization, M.K. and G.K.; methodology, M.K. and G.K.; formal analysis, A.A., Ş.Ş.O. and G.K.; investigation, A.A., Ş.Ş.O. and G.K.; writing—original draft preparation, A.A. and Ş.Ş.O.; writing—review and editing, M.K. and G.K.; supervision, G.K.; project administration, G.K. All authors have read and agreed to the published version of the manuscript.

**Funding:** This study has been supported by The Scientific and Technological Research Council of Turkey (TÜBİTAK, Project number: 107O769).

**Institutional Review Board Statement:** Not applicable.

**Informed Consent Statement:** Not applicable.

**Data Availability Statement:** Not applicable.

**Conflicts of Interest:** The authors declare no conflict of interest.

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
