# Peer review of "Microbiological, Physicochemical and Sensorial Changes during the Ripening of Sucuk, a Traditional Turkish Dry-Fermented Sausage: Effects of Autochthonous Strains, Sheep Tail Fat and Ripening Rate"

_fermentation, doi:10.3390/fermentation9060558_

Round 1

Reviewer 1 Report

This study emphasizes importance of the influence of autochthonous starter strains, ripening rate and sheep tail fat on product properties of sucuk, a Turkish traditional dry fermented sausage, during ripening. Microbiological and physicochemical parameters (pH, water activity, TBARS- an indicator of lipid oxidation, and color) were performed on sucuk samples on certain days of ripening. Likewise, the sensory properties of mature sausages were examined. Introduction clearly describes background and defines scope of the investigation. The results and discussion appropriately interpret the significance of the research findings.

I suggest the following corrections to the manuscript:

1.  Lines 179-466: 2. Results and Discussion. This section should be numbered as 3.

2.  Line 312: Figure 5b should be Figure 3b.

3. Lines 319 and 340: The legends of figures 3 and 4 overlap with the graphics.

4. Line 466: 5. Conclusions. This section should be numbered as 4.

5.  In the conclusion, the main findings of the study are briefly stated and repeated. But future research plans regarding this product should also be emphasized. For example: Is there enough data on its nutritional value and biological properties, possible cytotoxicity..... It should also be pointed out how important research is for placing this product on the world market compared to similar products in other countries.

Author Response

Dear Reviewer,

Thank you for your valuable contributions and comments.

Kind regards,

Reviewer 1

This study emphasizes importance of the influence of autochthonous starter strains, ripening rate and sheep tail fat on product properties of sucuk, a Turkish traditional dry fermented sausage, during ripening. Microbiological and physicochemical parameters (pH, water activity, TBARS- an indicator of lipid oxidation, and color) were performed on sucuk samples on certain days of ripening. Likewise, the sensory properties of mature sausages were examined. Introduction clearly describes background and defines scope of the investigation. The results and discussion appropriately interpret the significance of the research findings.

Thank you for your comments.

I suggest the following corrections to the manuscript:

  1. Lines 179-466: 2. Results and Discussion. This section should be numbered as 3.

-Changed

  1. Line 312: Figure 5b should be Figure 3b.

-Changed

  1. Lines 319 and 340: The legends of figures 3 and 4 overlap with the graphics.

-Changed

  1. Line 466: 5. Conclusions. This section should be numbered as 4.

-Changed

  1. In the conclusion, the main findings of the study are briefly stated and repeated. But future research plans regarding this product should also be emphasized. For example: Is there enough data on its nutritional value and biological properties, possible cytotoxicity..... It should also be pointed out how important research is for placing this product on the world market compared to similar products in other countries.

-Revised: On the other hand, it is also important to determine the effects of autochthonous strains on the volatile compounds and textural parameters of sucuk under different process conditions. In addition, the use of starter culture is essential for a standard production in industrial production. In this respect, it is of great importance to transform autochthonous strains into commercial preparations.

Reviewer 2 Report

The study aimed to investigate the effects of autochthonous starter cultures (spontaneous fermentation, Lactiplantibacillus plantarum GM77, Staphylococcus xylosus GM92 or L. plantarum GM77 + S. xylosus GM92) isolated form sucuk (a Turkish traditional dry fermented sausage), the use of sheep tail fat (beef fat-control, sheep tail fat and beef fat + sheep tail fat) and ripening rate (slow or fast) on the microbiological, physicochemical and sensory changes during the ripening of sucuk. The experimental design of this study is reasonable and innovative, making it an interesting study. It is recommended that the manuscript be revised and accepted for publication.The specific modifications are as follows: 

1. Line 20: form?

2. Line 27: under?

3. Line 43-45: Microorganisms that may adversely affect product properties, which? Some pathogenic microorganisms, as well as technologically important microorganisms during ripening, which? Please elaborate!

4. Line 48-49: How do lactic acid bacteria inhibit spoilage and pathogenic bacteria?

5. Line 54: Why does the amount of fat decrease the pH during ripening?

6. Line 89-91: How to sample and sample weight? Quantity?

7. Line 113-119: What is the basis of the fermentation temperature setting? Why do you set it that way?

8. Line 123-125: After 0, 1, 3, 7, 9, 11, and 14 days, 2 samples were randomly selected from each group for microbial and physicochemical detection. Is it parallel? Can two samples theoretically explain whether the data are reliable and reasonable? Does not reflect the validity of the test data.

9. Line 129: Please delete “and”.

10. Line 130: transfered?

11. Line 131: were?

12. Line 149: using a the colorimeter device? Please rewrite this sentence.

13. Line 152: Were members trained before sensory evaluation? If you have trained, please write down the specific training method. Or are they experienced themselves?

14. Line 171: Two replicates cannot explain the problem, please supplement the test and data.

15. Line 173: 'P' instead of 'P'

16. Line 177: Are other processes, such as boiling or otherwise, performed during the sensory evaluation of the final product?

17. Line 179, 181, 238, 286, 294, 322, 342, 371, 424, 466: Please change the title number.

18. Line 215-216: 'P' instead of 'p'

19. Line 235: As can be seen from Figure 1 (a), the standard deviations of Ripening time 0 and 1 day are too large, the data are unreasonable, and the overall chart of the paper is not beautiful enough.

20. Line 212: Please make the table in a three-line format.

21. Line 235: As can be seen from Figure 1 (a), the standard deviations of Ripening time 0 and 1 day are too large, the data are unreasonable, and the overall chart of the paper is not beautiful enough.

22. Line 243: previously?

23. Line 249-251: Why did the pH decline slowly in the control group and in the group containing L. plantarumGM77?

24. Line 251: was?

25. Line 283: Please redraw - Figure 2 (c).

26. Line 299: The lowest pH value in fast ripening was determined on the 7th day, please explain the reasons.

27. Line 300: in the 3rd?

28. Line 318-321: The title of the image is blocked, please change it.

29. Line 337-340: The title of the image is blocked, please change it. The vertical title of the figure is blocked, please change it.

30. Line 351: are?

31. Line 368: Please standardize the legend format of all the figures in your article.

32. Line 384: Where is Table 2?

33. Line 365-370: Figure 5 (b) is blocked, please modify.

34. Line 418-419: Please change the title of the table.

35. Line 420-421: 'P' instead of 'p'

36. Line 453: Note the formatting issue of salience.

37. Line 463: Please redo Figure 7 without wrapping the contents of the horizontal coordinate.

38. Line 466: 'Conclusion' instead of 'Conclusions'

39. Line 494: Please add the DOI of the reference.

40. Re-check every single line from here to the end, since deficiencies are really numerous.

The language should be revised and highly improved. There are many grammar, format and language inconsistencies in the document which I have not pointed them.

Author Response

Dear Reviewer,

Thank you for your valuable contributions and comments.

Kind regards,

Reviewer 2

The study aimed to investigate the effects of autochthonous starter cultures (spontaneous fermentation, Lactiplantibacillus plantarum GM77, Staphylococcus xylosus GM92 or L. plantarum GM77 + S. xylosus GM92) isolated form sucuk (a Turkish traditional dry fermented sausage), the use of sheep tail fat (beef fat-control, sheep tail fat and beef fat + sheep tail fat) and ripening rate (slow or fast) on the microbiological, physicochemical and sensory changes during the ripening of sucuk. The experimental design of this study is reasonable and innovative, making it an interesting study. It is recommended that the manuscript be revised and accepted for publication.The specific modifications are as follows: 

  1. Line 20: form?

it was changed as "from"

  1. Line 27: under?

Changed

  1. Line 43-45: Microorganisms that may adversely affect product properties, which? Some pathogenic microorganisms, as well as technologically important microorganisms during ripening, which? Please elaborate!

-Added: This microbiota, known as house flora, may contain microorganisms such as Enterobacteriaceae, Pseudomonas and Enterococcus that may adversely affect product properties and some pathogenic microorganisms such as Staphylococcus aureus and Listeria monocytogenes, as well as technologically important microorganisms during ripening

  1. Line 48-49: How do lactic acid bacteria inhibit spoilage and pathogenic bacteria?

-revised: Generally, lactic acid bacteria are used to ensure acidification, and thanks to this acidification, lactic acid bacteria show an inhibitory effect against spoilage microbiota and pathogens

  1. Line 54: Why does the amount of fat decrease the pH during ripening?

-Thank you. Revised.

  1. Line 89-91: How to sample and sample weight? Quantity?

- Revised: For each treatment, 4 kg of beef + 1 kg of fat were used. Two productions were made for each treatment; thus 48 batters were produced according to the experimental design (given in supplementary material 1).

  1. Line 113-119: What is the basis of the fermentation temperature setting? Why do you set it that way?

-Generally, two different initial fermentation temperatures are applied in sucuk production. Therefore, these temperatures were chosen (18°C -slow  and 24°C- fast). In the production of fermented sausages, temperature and relative humidity are reduced depending on time. Therefore, such a program was used.

  1. Line 123-125: After 0, 1, 3, 7, 9, 11, and 14 days, 2 samples were randomly selected from each group for microbial and physicochemical detection. Is it parallel? Can two samples theoretically explain whether the data are reliable and reasonable? Does not reflect the validity of the test data.

-For each production, two samples were taken from each treatment. The analyzes were carried out on samples taken after 0, 1, 3, 7, 9, 11 and 14 days of ripening. The sensory assessment was also carried out on the 14th day of ripening (final product).

  1. Line 129: Please delete “and”.

Deleted

  1. Line 130: transfered?

Corrected

  1. Line 131: were?

Corrected

  1. Line 149: using a the colorimeter device? Please rewrite this sentence.

Revised: The color values (L*(lightness), a* (red-green component) and b* (yellow-blue component)) in samples sliced in 1 cm thickness were measured using a Chroma meter (Minolta, Osaka, Japan) with D65 illuminant, an aperture size of 8 mm and 2° standard observer. Measurements were performed in triplicate. Calibration was performed on a standard white plate (Y = 87.4, x = 0.3184, y = 0.3364) to standardize the equipment before each measurement

  1. Line 152: Were members trained before sensory evaluation? If you have trained, please write down the specific training method. Or are they experienced themselves?

-The sensorial panels are divided into three categories: trained, semi-trained and consumer panel. Trained and semi-trained categories are used in the laboratory conditions for research and development work. In addition, this type of panel cosists of people who are usually familiar with quality of foods. This panel is capable of discriminating differences and communicating their reactions, although they may not have been formally trained. The panelist number consists of about 8 to 25 members.

  1. Line 171: Two replicates cannot explain the problem, please supplement the test and data.

Revised: The experiment was carried out according to a completely randomized design with two replicates (two batters for each treatment). Two independent batters of sausages (replicates) were prepared on different days. The ripening rate, sheep tail fat, starter culture and ripening time were evaluated as main effects, and replications were evaluated as random effects. The analysis of variance was applied to the data. The means of significant sources of variation were compared using Duncan's multiple range tests at the p < 0.05 level. The results were expressed as the mean values ± standard deviation (Sd). All statistical analysis was performed using the SPSS statistical program (IBM SPSS Inc., Chicago, IL, USA).

  1. Line 173: 'P' instead of 'P'

It was revised as ”p”

  1. Line 177: Are other processes, such as boiling or otherwise, performed during the sensory evaluation of the final product?

Sucuk can be consumed both raw and cooked. In this study, sensory analysis was performed on raw samples, as the cooking  obscures the product's distinctive properties.

  1. Line 179, 181, 238, 286, 294, 322, 342, 371, 424, 466: Please change the title number.

Changed.

  1. Line 215-216: 'P' instead of 'p'

 Changed.

  1. Line 235: As can be seen from Figure 1 (a), the standard deviations of Ripening time 0 and 1 day are too large, the data are unreasonable, and the overall chart of the paper is not beautiful enough.

-As can be seen from the N values (both for N= 144) of slow and fast ripening given in Table 1, it is quite normal that the deviations are around 2 log units. Because statistics were created over 144 values for each ripening rate. The this figure  only shows the interaction of ripening rate and ripening time.In the calculation of these averages, the averages of both the groups with and without the starter culture were used. The number of lactic acid bacteri in sucuk batters is around 4 log cfu/g in groups without starter culture, and around 7 log cfu/g in samples with starter culture.

  1. Line 212: Please make the table in a three-line format.

-Revised.

  1. Line 235: As can be seen from Figure 1 (a), the standard deviations of Ripening time 0 and 1 day are too large, the data are unreasonable, and the overall chart of the paper is not beautiful enough.

This is the same as question 19. The answer is given in no 19.

  1. Line 243: previously?

Changed

  1. Line 249-251: Why did the pH decline slowly in the control group and in the group containing L. plantarumGM77?

The sentence has been revised as follows:

As can be seen from Table 1, spontaneous micrococci and staphylococci did not grow well in the group containing L. plantarum GM77 due to the rapid drop in pH value.

  1. Line 251: was?

-Revised

  1. Line 283: Please redraw - Figure 2 (c).

-Revised.

  1. Line 299: The lowest pH value in fast ripening was determined on the 7th day, please explain the reasons.

Revised: As seen in Figure 3a, in rapid ripening, the pH started to decrease after 24 hours of fermentation and this decrease continued until the 7th day. The most significant decrease was observed on the 3rd day of ripening

  1. Line 300: in the 3rd?

Revised.

  1. Line 318-321: The title of the image is blocked, please change it.

Revised.

  1. Line 337-340: The title of the image is blocked, please change it. The vertical title of the figure is blocked, please change it.

Revised.

  1. Line 351: are?

Changed.

  1. Line 368: Please standardize the legend format of all the figures in your article.

Revised.

  1. Line 384: Where is Table 2?

Revised.

  1. Line 365-370: Figure 5 (b) is blocked, please modify.

Revised.

  1. Line 418-419: Please change the title of the table.

Revised.

  1. Line 420-421: 'P' instead of 'p'

Revised.

  1. Line 453: Note the formatting issue of salience.

Revised.

  1. Line 463: Please redo Figure 7 without wrapping the contents of the horizontal coordinate.

Revised.

  1. Line 466: 'Conclusion' instead of 'Conclusions'

Revised.

  1. Line 494: Please add the DOI of the reference.

-Added.

  1. Re-check every single line from here to the end, since deficiencies are really numerous.

-Checked

The language should be revised and highly improved. There are many grammar, format and language inconsistencies in the document which I have not pointed them.

-Checked

Reviewer 3 Report

The submitted manuscript is original and it does expand the knowledge in the area of dry-fermented sausages. We believe it significantly builds on (the author's) previous work and fits the scope of the journal. The writing style is clear and appropriate and tables and figures are clear to read and labeled appropriately.

However, there are some MAJOR issues that need to be resolved before publication:

#1 The need for a standardized set of minimum reportable parameters for instrumental meat color evaluation still remains to be identified and incorporated in peer-reviewed journal guidelines for authors, as it was the case a decade ago. In the most recent review regarding meat color https://doi.org/10.1016/j.cofs.2021.02.012  the authors are proposing that all manuscripts containing instrumental color data must report instrumental details that include (at least) the information on: the instrument and its calibration, illuminant, aperture size, degree of observer and a number of readings per sample. Unfortunately, this manuscript missed reporting some of these parameters as well. Please correct this.

#2 The magnitude of color difference between the two methods used is best represented by the total color difference value (ΔE). The info about how it is calculated and about its threshold for human meat-color difference detection is described in https://doi.org/10.1016/j.meatsci.2018.09.015    Therefore, please calculate ΔE for your results when comparing it to the control sample and discuss the l,a,b values you have obtained with the values for pork sausage in the work of https://doi.org/10.1016/j.meatsci.2018.09.015

#3 When setting up a sensory panel, the first step is its validation to confirm that the panel can work and be used in sensory studies. Also, validation of training (prior to sensory analysis), may result in the exclusion of panelists due to discriminating problems. No information is provided regarding validation methods used and resulting activities applying to this sensory panel (https://doi.org/10.1111/jtxs.12616 ).

#4 Besides validating the panel (prior to sensory analysis), it is also important to assess panelists’ performance. Criteria for evaluating the attributes of a trained sensory panel and evaluation of the panel performance cover the following aspects: (i) is the panel capable of showing products differences / discriminate in-between samples; (ii) are the scores of panelists reliable (in-between replicates and over time in case of evaluating products over time); (iii) are results valid in terms of visible consensus between panelists and scoring in a similar way; and (iv) are they able to specify specific sensory attributes and sensations. Standard ISO 11132 outlines all four criteria as of equal importance: discriminability (linked with differences between products), homogeneity (consensus of the panel), repeatability (within sessions), and reproducibility (between-session) as well as two-way ANOVA for panel performance (discrimination, homogeneity and repeatability) and one-way ANOVA for assessor performance (discrimination and repeatability). No information is provided regarding the assessment of the panel performance used in this study. Without the information on sensory panel validation and assessment of the panel performance, we cannot trust the results obtained by the panel.

#5 Finally, and most importantly, the Hedonic sensory test used is not designed for trained panels but for consumers. Wide consensus exists regarding the idea that trained assessors cannot perform hedonic tests, as they are trained to leave out their personal preferences and to evaluate products using specific criteria. Added to this, a small trained panel (usually n 10) could never be representative of a target market (Stone & Sidel, 2004). Thus, hedonic perception of products by a few trained assessors does not represent naïve consumers wide and varied perception and cannot be regarded as a measure of the potential performance of the product in the marketplace (Lawless and Heymann, 2010, O’Mahony, 1979). https://doi.org/10.1016/j.foodqual.2016.10.006 Therefore, the authors need to make a different sensory test and report the results back or use the same test but direct it to couple of hundreds of consumers.

Minor editing of English language required.

Author Response

Dear Reviewer,

Thank you for your valuable contributions and comments.

Kind regards,

Reviewer 3

The submitted manuscript is original and it does expand the knowledge in the area of dry-fermented sausages. We believe it significantly builds on (the author's) previous work and fits the scope of the journal. The writing style is clear and appropriate and tables and figures are clear to read and labeled appropriately.

However, there are some MAJOR issues that need to be resolved before publication:

#1 The need for a standardized set of minimum reportable parameters for instrumental meat color evaluation still remains to be identified and incorporated in peer-reviewed journal guidelines for authors, as it was the case a decade ago. In the most recent review regarding meat color https://doi.org/10.1016/j.cofs.2021.02.012  the authors are proposing that all manuscripts containing instrumental color data must report instrumental details that include (at least) the information on: the instrument and its calibration, illuminant, aperture size, degree of observer and a number of readings per sample. Unfortunately, this manuscript missed reporting some of these parameters as well. Please correct this.

The color values (L*(lightness), a* (red-green component) and b* (yellow-blue component)) in samples sliced in 1 cm thickness were measured using a Chroma meter (Minolta, Osaka, Japan) with D65 illuminant, an aperture size of 8 mm and 2° standard observer. The measurements were performed in triplicate. Calibration was performed on a white standard plate (Y = 87.4, x = 0.3184, y = 0.3364) to standardize the equipment before each measurement.

#2 The magnitude of color difference between the two methods used is best represented by the total color difference value (ΔE). The info about how it is calculated and about its threshold for human meat-color difference detection is described in https://doi.org/10.1016/j.meatsci.2018.09.015    Therefore, please calculate ΔE for your results when comparing it to the control sample and discuss the l,a,b values you have obtained with the values for pork sausage in the work of https://doi.org/10.1016/j.meatsci.2018.09.015

In this study, color measurement was carried out with only one method. In other words, two different methods were not used. In the study you mentioned, "Comparison of a computer vision system vs. traditional colorimeter (https://doi.org/10.1016/j.meatsci.2018.09.015)" was made. From this point of view, it does not seem possible to make this evaluation here. However, I checked the work you cited and it changed my perspective on color evaluation. In some other studies, I realized, there were researchers who calculated the color difference value (ΔE) based on the control. However, since there are 4 factors in this study, it does not seem possible to make such a comparison. Thanks for your contributions.

#3 When setting up a sensory panel, the first step is its validation to confirm that the panel can work and be used in sensory studies. Also, validation of training (prior to sensory analysis), may result in the exclusion of panelists due to discriminating problems. No information is provided regarding validation methods used and resulting activities applying to this sensory panel (https://doi.org/10.1111/jtxs.12616 ).

As you known, the sensorial panels are divided into three categories: trained, semi-trained and consumer panel. Trained and semi-trained categories are used in the laboratory conditions for research and development work. In addition, this type of panel cosists of people who are usually familiar with quality of foods. This panel is capable of discriminating differences and communicating their reactions, although they may not have been formally trained. The panelist number consists of about 8 to 25 members (Amerine, M.A., Pangborn, R.M. and Roessler, E.B. (1965). Principles of Sensory Evaluation of Food, Academic Press, New York).

In this study, 20 semi-trained panelists participated in the sensory evaluation.

#4 Besides validating the panel (prior to sensory analysis), it is also important to assess panelists’ performance. Criteria for evaluating the attributes of a trained sensory panel and evaluation of the panel performance cover the following aspects: (i) is the panel capable of showing products differences / discriminate in-between samples; (ii) are the scores of panelists reliable (in-between replicates and over time in case of evaluating products over time); (iii) are results valid in terms of visible consensus between panelists and scoring in a similar way; and (iv) are they able to specify specific sensory attributes and sensations. Standard ISO 11132 outlines all four criteria as of equal importance: discriminability (linked with differences between products), homogeneity (consensus of the panel), repeatability (within sessions), and reproducibility (between-session) as well as two-way ANOVA for panel performance (discrimination, homogeneity and repeatability) and one-way ANOVA for assessor performance (discrimination and repeatability). No information is provided regarding the assessment of the panel performance used in this study. Without the information on sensory panel validation and assessment of the panel performance, we cannot trust the results obtained by the panel.

- In the study, sensory analysis was carried out by semi-trained panelists. These panelists were not considered trained panelists. This panel consisted of members who are normally familiar with different classes of foods. Specific characteristics or differences of the product were evaluated. In this type evaluation, the panelist number consists of about 8 to 25 members (Amerine, M.A., Pangborn, R.M. and Roessler, E.B. (1965). Principles of Sensory Evaluation of Food, Academic Press, New York)

#5 Finally, and most importantly, the Hedonic sensory test used is not designed for trained panels but for consumers. Wide consensus exists regarding the idea that trained assessors cannot perform hedonic tests, as they are trained to leave out their personal preferences and to evaluate products using specific criteria. Added to this, a small trained panel (usually n ≅ 10) could never be representative of a target market (Stone & Sidel, 2004). Thus, hedonic perception of products by a few trained assessors does not represent naïve consumers’ wide and varied perception and cannot be regarded as a measure of the potential performance of the product in the marketplace (Lawless and Heymann, 2010, O’Mahony, 1979). https://doi.org/10.1016/j.foodqual.2016.10.006 Therefore, the authors need to make a different sensory test and report the results back or use the same test but direct it to couple of hundreds of consumers.

-In the study,  hedonic type scale, the most commonly used evaluation technique for measuring acceptability or preference, was used. As you mentioned, although there is a generalization about the idea that trained assessors cannot perform hedonic tests, the hedonic type scale was used by trained panelists in many studies. The doi of some related examples are listed below.

The reasons for preferring the structured-numbered hedonic type scale in this study are: It allows us to convert the response into a numeric value, for example 1=dislike very much – 9=extremely like it, and provides quick information about capacity and potential for success of the newly developed product.

https://doi.org/10.1111/j.1750-3841.2011.02497.x

 https://doi.org/10.1016/j.fbp.2013.03.001

https://doi.org/10.1007/s13197-019-03754-1

https://doi.org/10.1016/j.lwt.2014.04.003

https://doi.org/10.1016/j.profoo.2016.02.033

DOI: 10.4236/fns.2019.1011096